# Inhibition of Antiestrogen-Promoted Pro-Survival Autophagy and Tamoxifen Resistance in Breast Cancer through Vitamin D Receptor

**DOI:** 10.3390/nu13051715

**Published:** 2021-05-19

**Authors:** Ye Li, Katherine L Cook, Wei Yu, Lu Jin, Kerrie B Bouker, Robert Clarke, Leena Hilakivi-Clarke

**Affiliations:** Department of Oncology, Georgetown University, Washington, DC 20057, USA; Ye.Li@uth.tmc.edu (Y.L.); klcook@wakehealth.edu (K.L.C.); yuwei30@gmail.com (W.Y.); lujin@umn.edu (L.J.); Kerrie.Bouker@georgetown.edu (K.B.B.); clarker@umn.edu (R.C.)

**Keywords:** breast cancer, recurrence-free survival, tamoxifen, vitamin D receptor, vitamin D analogs, autophagy

## Abstract

We determined how vitamin D receptor (VDR) is linked to disease outcome in estrogen receptor-positive (ER+) breast cancer patients treated with tamoxifen (TAM). Breast cancer patients (*n* = 581) in four different datasets were divided into those expressing higher (above median) and lower levels of VDR in pretreatment ER+ tumors. Across all datasets, TAM-treated patients with higher pretreatment tumor VDR expression exhibited significantly longer recurrence-free survival. Ingenuity pathway analysis identified autophagy and unfolded protein response (UPR) as top differentially expressed pathways between high and low VDR-expressing ER+ cancers. Activation of VDR with vitamin D (VitD), either calcitriol or its synthetic analog EB1089, sensitized MCF-7-derived, antiestrogen-resistant LCC9 human breast cancer cells to TAM, and attenuated increased UPR and pro-survival autophagy. Silencing of VDR blocked these effects through the IRE1α-JNK pathway. Further, silencing of VDR impaired sensitivity to TAM in antiestrogen-responsive LCC1 cells, and prevented the effects of calcitriol and EB1089 on UPR and autophagy. In a preclinical mouse model, dietary VitD supplementation induced VDR activation and reduced carcinogen-induced ER+ mammary tumor incidence. In addition, IRE1α-JNK signaling was downregulated and survival autophagy was inhibited in mammary tumors of VitD-supplemented mice. Thus, activation of VDR is predictive of reduced risk of breast cancer recurrence in ER+ patients, possibly by inhibiting antiestrogen-promoted pro-survival autophagy.

## 1. Introduction

In 2020, breast cancer surpassed lung cancer as the most commonly diagnosed cancer worldwide, with approximately 2.3 million women diagnosed with this disease [1]. Over 70% of breast cancers are estrogen receptor positive (ER+) [2,3], and ER activation is associated with breast cancer development and progression. Targeted therapies that prevent ER activation, such as use of the partial ER antagonist tamoxifen (TAM) in premenopausal patients, are effective in treating breast cancer [4]. However, a major clinical challenge is that although the 5-year survival rate for invasive breast cancer is 90%, over 50% of those with non-metastatic, lymph node-positive, ER+ breast cancer at diagnosis will recur within 20 years of diagnosis [5]. Understanding the biological causes of antiestrogen resistance, and identifying effective means to prevent and reverse resistance, will lead to significant reductions in breast cancer mortality.

The unfolded protein response (UPR) is associated with an acquired endocrine-resistant phenotype [6]. UPR is activated in normal cells under conditions such as oxidative stress, nutrient deprivation, or hypoxia, leading to the accumulation of unfolded or misfolded proteins in the lumen of the endoplasmic reticulum [7]. In the initial phase of UPR, activation of inositol-requiring enzyme-1 (IRE-1, *ERN1*), protein kinase RNA-like endoplasmic reticulum kinase (PERK, *EIF2AK3*), and activating transcription factor 6 (ATF6) can resolve the stress by slowing down protein accumulation and improving folding capacity [8]. In cancer cells, UPR remains chronically activated due to constant stress. One of the UPR-regulated downstream pathways is autophagy, a “self-eating” process that occurs in all cells to carry out the proper degradation of proteins, protein aggregates, or damaged organelles. Cancer cells re-use the products of degradation, allowing them to survive in a hostile cellular environment [9]. Increased pro-survival autophagy is causally linked to endocrine resistance in breast cancer [10,11]. However, autophagy can also induce cell death and eliminate cancer cells, and thus autophagy is called a “double-edged sword” [12]. For example, autophagy inhibits cancer development but once tumors are present, autophagy helps cancer cells to survive and grow [13].

Activation of IRE-1α and its downstream target Jun N-terminal kinase (JNK) can induce autophagy by upregulating Beclin 1 and phosphorylating B-cell lymphoma 2 (Bcl-2) [9]. Beclin 1 does not activate downstream autophagy events when it is bound to non-phosphorylated Bcl-2 [14]. However, phosphorylated Bcl-2 disrupts the Bcl-2/Beclin1 complex, releasing Beclin 1 and enabling autophagy [15,16]. Beclin 1 forms a scaffold for recruitment of autophagy-related genes (ATGs) to initiate autophagosome formation. ATGs convert LC3 (soluble LC3-I) to lipid bound LC3-II [17], which plays a central role in autophagosomal membrane elongation and cargo selection. Autophagosomes fuse with lysosomes to form autolysosomes, in which p62 (*SQSTM1*), an ubiquitin-/LC3-binding protein, labels cargo proteins and is then degraded as part of autophagic flux [18]. 

Vitamin D (VitD) initiates autophagy in breast cancer cells [19,20,21]. Vitamin D_3_ (VD_3_, cholecalciferol) is obtained by synthesis in the skin through ultraviolet exposure and dietary intake of VD_3_-containing or supplemented foods. VD_3_ is converted to 25(OH)D_3_ in the kidney and further to 1,25(OH)2D_3_ (calcitrol) in the liver. Calcitriol is a steroid hormone that binds and activates the vitamin D receptor (VDR), and plays a central role in calcium homeostasis and in regulating cell growth and proliferation [22,23]. The vitamin D analog, EB1089, has regulatory effects on VDR similar to calcitriol, but less adverse effects on calcium metabolism [24,25]. In earlier studies, 100 nM calcitriol and 100 nM EB1089 were found to efficiently induce pro-death autophagy in MCF-7 human breast cancer cells and to inhibit growth of these cells [26,27].

Since VitD upregulates VDR expression [28,29,30], and VDR is known to repress autophagy in human breast cancer cells and in mice [26], we hypothesized that VitD can inhibit the pro-survival autophagy that is initiated by treating antiestrogen-resistant breast cancer cells with tamoxifen (TAM) [10,11].

We show here that ER+ breast cancer patients with a higher than average level of VDR expression in their pretreatment tumor survived significantly longer than patients with lower tumor VDR expression. In addition, autophagy signaling pathways were suppressed in breast tumors exhibiting high levels of VDR. In an in vivo model, VD3 supplementation inhibited mammary cancer development and tumor autophagy in mice. In vitro studies revealed that both calcitriol and EB1089 inhibited TAM-induced upregulation of IRE1α, JNK, Beclin 1, ATG7, LC3BII, and pBcl-2 and increased p62 levels in MCF-7-derived, TAM-resistant LCC9 human breast cancer cells. VDR agonists also sensitized LCC9 cells to TAM-induced inhibition of cell growth, and the effect was blocked by silencing VDR. Thus, VitD may reverse antiestrogen resistance by inhibiting pro-survival autophagy.

## 2. Materials and Methods

**VDR levels and recurrence-free survival in ER+ breast cancer patients.** We determined whether VDR levels are associated with relapse-free survival among ER+ breast cancer patients. Recurrence or relapse-free survival (RFS) in human datasets was defined as the interval from breast surgery to the diagnosis of the first local or distant recurrence. We used two inclusion criteria for our choice of datasets: (1) histological verification of tumor ER status, and (2) patients receiving post-surgical adjuvant TAM therapy. Four independent datasets fulfilling these criteria were included in the analysis: GEO profile accessions = GSE-17705 (*n* = 298 cases, 0 missing values, median follow-up time: 84.95 months) [31], GSE-12093 (*n* = 136 cases, 0 missing values, median follow-up time: 97.75 months) [32], GSE-6532 (*n* = 87 cases, 4 missing values, median follow-up time: 136.24 months) [33], and GSE-1379 (*n* = 60 cases, 4 missing values, median follow-up time: 88.50 months) [34]. Brief descriptions of the datasets are provided in Appendix A. For each GSE dataset, survival analysis was performed on patients with high (>median) versus low (≦median) VDR expression.

**Identification of differentially expressed genes in ER+ tumors expressing high or low VDR.** In eachs of the four datasets, we identified differentially expressed genes (DEGs) with FDR < 0.05 between tumors expressing high or low VDR. Enrichment analysis of DEGs between high and low VDR-expressing tumors was performed by Ingenuity Pathway Analysis (IPA, Qiagen, Redwood City, CA, USA). We also determined potential overlap between DEGs and autophagy genes in the Autophagy Regulator Network (ARN) developed by Turei et al. to be used as a bioinformatics resource for studying the mechanisms and regulation of autophagy [35].

**In vitro TAM resistance model.** We used our well-characterized MCF-7-derived in vitro model of TAM-responsive (LCC1) or -resistant (LCC9) ER+ human breast cancer [36,37]. Details of the authentication of the cell lines are provided in Appendix B. Both cell lines express the VDR (LCC1 and LCC9 cells were grown in phenol red-free IMEM media containing 5% charcoal stripped calf serum. Cells were grown at 37 °C in a humidified, 5% CO_2_:95% air atmosphere. Cells were treated with 100 nM of the VDR agonists calcitriol or EB1089. Doses of agonists were chosen to reflect previous studies that investigated the link between VitD and autophagy [19,20,21,26]. 

**siRNA Transfection.** LCC1 and LCC9 cells were plated at 1 × 10^5^ cells/well on a 24 well tissue culture plate. 10 nM of three unique 27mer VDR siRNA duplexes and control siRNA (Origene) were transfected using Lipofectamine RNAiMax transfection reagent (Invitrogen-Life Technologies, Inc., Carlsbad, CA, USA).

**Crystal violet cell density assays.** First, 24 h post-siRNA transfection, LCC1 and LCC9 cells were treated with varying doses (0–1000 nM) of 4-hydroxy TAM (Sigma-Aldrich, St. Louis, MO, USA) with or without 100 nM calcitriol (Tocris) or 100 nM EB1089 (Tocris). Three days after treatment, the culture medium was removed by aspiration; cells were washed with PBS, stained with crystal violet (Fisher Scientific, Fairlawn, NJ, USA) for 20 min, washed to remove excess stain and dried overnight. Stained cells were permeabilized using citrate buffer and light absorbance was read at 480 nm on a plate reader for quantitative analysis. 

**Western blot analyses.** Cells were lysed in RIPA buffer (50 mmol/L Tris-HCl, pH 7.4, 150 mmol/L NaCl, 1% NP40, 0.25% Na-deoxycholate) containing 1× cOmplete^TM^ Mini protease inhibitor cocktail (Roche Diagnostics, Basel, Switzerland) and 1× PhosStop phosphatase inhibitor cocktail (Roche Diagnostics, Basel, Switzerland) tablets on ice and sonicated. Proteins were measured by a standard bicinchoninic acid assay, size fractioned by polyacrylamide gel electrophoresis (PAGE), and transferred to a nitrocellulose membrane. Non-specific binding was blocked by incubation for 1 h in 5% powdered milk in tris-buffered saline containing Tween-20 (TBST-Milk) plus 1% Triton X-100 solution, followed by incubation with primary antibody overnight at 4 °C. Antibodies used were IRE1, pIRE1, JNK, pJNK, pBcl-2, Bcl-xL, Beclin 1, ATG7, ATG 5, LC-3B, and VDR (1:200 to 1:500) (all purchased from Cell Signaling Technology, New England Biolabs, Ipswich, MA, USA), p62 (BD Biosciences, San Diego, CA, USA), and Bcl-2 (Enzo Life Sciences, Farmingdale, NY, USA). Species-specific polyclonal horseradish peroxidase (HRP)-conjugated secondary antibodies (1:2000) were used to incubate membranes for 1 h at room temperature. Immunoreactive products were visualized by chemiluminescence (SuperSignal Femto West; Pierce Biotechnology, Rockford, IL, USA). Protein expression was quantified by densitometry using ImageJ software (http://rsbweb.nih.gov/ij/ accessed on 9 November 2016). All samples were normalized to β-actin (1:1000; Santa Cruz Biotechnology, Santa Cruz, CA, USA). 

**In vivo experiments:** Female C57BL/6 mice were obtained from the Mouse Models for Human Cancer Consortium (MMHCC) at the National Cancer Institute (Frederick, MD, USA), and housed in the Department of Comparative Medicine facility at Georgetown University under a standard 12 h light–dark cycle. At 21 days of age, mice were weaned and female mice were divided into two groups, with 20 mice per group: (1) those fed a control AIN93G diet containing 1 international unit (IU) of VD_3_/g diet, and (2) those fed AIN93G diet supplemented with 25 IU VD_3_/g diet. VD3 supplementation in our study is similar to that used in a previous study showing that VDR is a major regulator of autophagy in mice [26]. Serum 25(OH)D_3_ levels were determined from both control and VD3-supplemented animals at 8 weeks of age using the 25(OH)D_3_ vitamin D ELISA assay test kit (Eagle Biosciences, Amherst, NH, USA) following the manufacturer’s instructions.

Mammary tumors in both control and VD3-supplemented animals were induced by priming mammary glands of 6-week-old mice with 15 mg medroxyprogesterone acetate (MPA) via intraperitoneal injection, and then administering 1 mg 7,12-dimethylbenz[a] anthracene (DMBA) in 0.1 mL cottonseed oil by oral gavage on postnatal week 7. DMBA was administered again on postnatal weeks 8, 9 and 10. Mice were examined for mammary tumors by palpation twice per week, and the latency of tumor appearance was assessed. Tumor sizes were measured using caliper once a week. Those animals in which tumor burden approximated 10% of total body weight were euthanized, as required by our institution. All other animals were euthanized 19 weeks after final carcinogen administration. Mammary tumors were removed at necropsy. A portion of each tumor was fixed in neutral-buffered 10% formalin, and processed for immunohistochemistry and histopathological analysis. All animal procedures were approved by the Georgetown University Animal Care and Use Committee, and the experiments were performed following the National Institutes of Health guidelines for the proper and humane use of animals in biomedical research.

**Immunohistochemistry of DMBA tumors:** Formalin-fixed mammary tumor sections were embedded in paraffin and cut into 5 mm sections. Immunostaining was performed with antibodies to VDR (1:100), Bcl-2 (1:100), pBcl-2 (1:100); p62 (1:1000), LC3 (1:100) using the streptavidin-biotin method. Antibody sources are listed in the Western blotting section above. Stained sections were visualized and photographed for scoring according to the Allred scoring system. 

**Statistical Analyses**: All data are shown as the mean ± SEM. Kaplan–Meier survival analysis with the Log Rank test was used to assess difference in RFS between patients with high or low VDR expression status within each GSE dataset. For in vitro experiments, differences between two groups were analyzed using student’s *t*-test. Multiple group comparisons were made by one-way ANOVA analysis followed by adjustments for multiple comparisons using Dunnett or Bonferroni post hoc tests, as appropriate. For in vivo experiments, differences in tumor incidence were analyzed using Kaplan–Meier analysis, followed by the Log Rank test. Analysis of autophagy genes and 25(OH)D_3_ levels were done by student’s t-test. Differences were considered to be statistically significant at *p* < 0.05. Statistical analysis was performed using Prism 5.0.

## 3. Results

### 3.1. High VDR Expression Is Associated with Longer Recurrence-Free Survival in TAM-Treated Breast Cancer Patients

In four independent databases consisting of a total of 581 ER+ breast cancer patients treated with TAM, higher pretreatment expression of VDR in the tumors was predictive of significantly longer RFS, compared with patients whose tumors expressed lower levels of VDR (Figure 1A–D). In each study, higher and lower VDR levels were determined based on median tumor VDR expression within that study. The median expression values were in the range of 6.2–6.7 when HG-U133A Affymetrix platform was used, 4.2 when HG-U133 Plus2.0 was used, and 1.3 when Acturus 22k human oligonucleotide microarray was used. The hazard ratios (HR) for RFS in the different datasets were the following: GSE-17005 dataset consisting of *n* = 298 eligible patients, HR was 0.444, [95% CI: 0.149–0.711] (Figure 1A), GSE-12093 consisting of *n* = 136 patients, HR was 0.307, [95% CI: 0.162–0.974] (Figure 1B), GSE-6532 consisting of *n* = 87 patients, HR was 0.375; [95% CI: 0.0757–0.797] (Figure 1C), and GSE-1379 consisting of *n* = 60 patients, HR was 0.434, [95% CI: 0.132–0.953] (Figure 1D). We also analyzed RFS in GSE-1456, a dataset of 159 cases of which only 39% were ER+ and treated with TAM [38]. No association between VDR expression and RFS was seen. These data indicate that higher VDR expression was predictive of longer RFS among TAM-treated ER+ breast cancer patients.

### 3.2. Differentially Expressed Genes (DEGs) in High versus Low VDR-Expressing ER+ Breast Cancers

We identified DEGs between tumors expressing high or low levels of VDR in each of the four datasets of TAM-treated patients (GSE-17705, GSE-12093, GSE-6532, and GSE-1379). DEGs (*p* < 0.05) from each dataset were then used to identify VDR-related signaling pathways by enrichment analysis using IPA (Appendix A). The results revealed the top three biological functions as cell death and survival, cellular proliferation, and cancer. The top biofunction cell death and survival contained 538 DEGs (Appendix A).

Among the top canonical pathways identified by IPA were autophagy and UPR, and other pathways closely linked to autophagy including HIF1α [39] and p53 signaling [40] (Appendix A). When DEGs in each dataset were overlaid with the Autophagy Regulatory Network (ARN), 30 DEGs linked to mechanism and regulation of autophagy were identified (Figure 2). Figure 1E,F shows the differential expression patterns of autophagy-related genes between patients with high and low VDR expression in two independent datasets (GSE-17705, GSE-12093). Several autophagy genes including Beclin 1, ATG4A, ATG5, ATG7, ATG10, and ATGF12 were downregulated in high-VDR tumors.

Interactions among the 30 differentially expressed autophagy genes are shown in Figure 2. ATG12, ESR1, BCL2, PIK3C3, and PRKAG2 were differentially expressed in all four datasets. ATG12, Bcl-2, and PIK3C3 are central autophagy regulators [41]. ESR1 can regulate each of these genes [11]. PRKAG2 is an AMPK family member; AMPK is a master regulator of cellular energy homeostasis and activates autophagy. Other VDR linked autophagy genes were ATG7, ULK2 (ATG1B), BECN1, ASPH, ANXA1, HAT1, PGK1, TKT, and GK1 that induce autophagy. Other DEGs included ATG3, ATG4A, ATG4B, ATG5, ATG10, ATG13, MAP1A, MAP1B, RABGAP1, and RB1CC1/ATG17 that are directly related to the process of autophagosome formation. Autophagy cargo receptor marker p62 (SQSTM1) was also differentially expressed. Moreover, some DEGs were AMPK family members (PRKAA1 and PRKAA2). Other DEGs identified from the Autophagy Regulatory Network included GLG1, OSBPL8, SEC23B, TKT and YLPM1. Taken together, these results show that VDR affects autophagy at multiple levels. 

### 3.3. Effects of Calcitriol and EB1089 on Cancer Cell Proliferation, UPR and Autophagy in LCC1 and LCC9 Human Breast Cancer Cell Lines

The effect of calcitriol and EB1089 as monotherapies on cell proliferation, UPR, and autophagy was studied in LCC1 (TAM-sensitive) and LCC9 (TAM-resistant) human ER+ breast cancer cell lines. Both calcitriol and EB1089 upregulated VDR in LCC1 and LCC9 cells (Appendix A). However, neither compound altered cell growth in a crystal violet assay within 72 h of treatment (Figure 3A), indicating no observed short-term effects on cell proliferation in LCC1 or LCC9 cells. In LCC9 cells, calcitriol and EB1089 downregulated IRE1α and pIRE1α (Figure 3B,C), JNK and phosphorylated JNK (Figure 3D,E) and phosphorylated Bcl-2 (Figure 3H), but total Bcl-2 levels were increased (Figure 3G). In LCC1 cells, total and phosphorylated JNK were also reduced and total Bcl-2 was increased. Since Beclin 1 cannot recruit key autophagy proteins to form a preautophagosomal structure when bound to Bcl-2, these data imply that calcitriol and EB1089 inhibit the initiation of autophagy. However, no changes in Beclin 1, ATG7, LC3II/LC3I, or p62 were seen by calcitriol or EB1089 in either LCC1 or LCC9 cells (Appendix A–F). Thus, as monotherapies, VitD compounds did not alter proliferation or autophagy of MCF-7-derived breast cancer cells.

### 3.4. VitD Restores TAM Sensitivity and Inhibits the UPR and Autophagy in Human Breast Cancer Cell Lines

Calcitriol and EB1089 restored sensitivity to TAM in LCC9 cells (Figure 4B). In LCC1 cells, both VitD compounds enhanced TAM’s ability to inhibit the relative cell proliferation index (Figure 4A). VDR levels were not increased by TAM alone. However, adding calcitriol or EB1089 significantly elevated VDR in both LCC1 and LCC9 cells (Appendix A).

Calcitiol and EB1089 modified changes in IRE1α-JNK caused by TAM treatment in LCC9 cells. TAM increased protein levels of IRE-1α, pIRE-1α/ IRE-1α, JNK and pJNK/JNK in LCC9 cells (Appendix A–E). Adding calcitriol or EB1089 reversed these TAM–induced changes in UPR-related genes in LCC9 cells (Appendix A). No changes by TAM or TAM+VitD were seen in these end-points in the LCC1 cells. 

TAM treatment reduced Bcl-2, and increased pBcl2/Bcl-2, Beclin 1, Atg7, and LC3BII in LCC9 cells indicative of autophagy (Figure 4D,F–I). Expression of these genes was also altered in LCC1 cells, although not as significantly as in LCC9 cells. Adding calcitriol or EB1089 reversed all of these changes in both cell lines. Further, calcitriol and EB 1079 significantly upregulated p62 expression in the TAM-treated LCC9 cells but not in LCC1 cells (Figure 4J). Thus, VitD inhibits TAM-induced autophagy and restores their sensitivity to TAM-induced inhibition of cell proliferation in TAM-resistant LCC9 cells.

### 3.5. Effects of VitD Are Mediated by VDR Expression

We next studied whether the effects of calcitriol and EB1089 on UPR and autophagy in LCC1 and LCC9 cells were dependent upon VDR. Silencing of VDR by siRNA led to a significant downregulation of VDR expression in LCC1 and LCC9 cells (Appendix A) but did not affect cell proliferation (Figure 5A). In both cell lines, VDR siRNA reduced total IRE1α (Figure 5C) and increased pIRE1α (Figure 5C), phosphorylated JNK, phosphorylated Bcl-2 (Figure 5H), LC3I (Appendix A) and LC3II expression (Appendix A). However, consistent with the findings that VitD compounds alone did not modify Beclin1, ATGF7, or p62 in LCC1 or LCC9 cells, knocking down VDR by siRNA also did not modify expression of these genes (Figure 5I and Appendix A).

In TAM-treated LCC1 cells, inhibiting VDR expression with siRNA prevented the TAM-induced inhibition of cell proliferation (Figure 6A). In LCC9 cells, calcitriol and EB1089 induced sensitization to TAM (Figure 6B). The changes in IRE1α, JNK, and Bcl-2 seen in cells not treated with TAM, where also seen when cells were treated with TAM. Thus, in the presence of TAM, reducing VDR expression by siRNA downregulated total IRE1α (Appendix A) and upregulated pIRE1α, pJNK, LC3I (Appendix A), pBcl-2 (Figure 6F), LC3II (Figure 6I), and SQSTM1/p62 expression (Figure 6J) in LCC9 cells. The increase in IRE1α, JNK, and LC3I, and LC3II expression was also seen in TAM-treated LCC1 cells. The effects of calcitriol and EB1083 in reversing TAM resistance, and in inhibiting TAM-induced pro-survival autophagy, were mediated by VDR because no changes were seen if VDR was eliminated by siRNA treatment. Appendix A summarizes all the data obtained using LCC1 and LCC9 cells.

### 3.6. VD_3_ Exposure Reduced Mammary Cancer Risk in Mice and Inhibited Survival Autophagy in Their Mammary Tumors

To determine whether VitD affects pro-survival autophagy in vivo, we studied its effects on mammary cancer risk and pro-autophagy markers in ER+ mammary tumors in mice. After weaning, mice were fed either a control diet containing nutritionally sufficient levels of VD_3_ or a VD_3_-supplemented diet. VD_3_ supplementation increased blood 25(OH)D_3_ levels by 2.1-fold (Figure 7A). The incidence of mammary cancers was significantly lower in mice fed the VD_3_-supplemented diet than in mice fed control diet (Figure 7B). Total tumor burden was also lower in mice fed high VD_3_-containing diet (Figure 7C). 

The IRE-1α/UPR pathway and autophagy markers were measured in the ER+ tumors of VD_3_-supplemented mice. VD_3_ supplementation increased IRE-1α and Bcl-2 and reduced pBcl-2 protein levels in the mammary tumors (Figure 7D–G). Autophagy markers Beclin 1, Atg5, and LC3B were significantly reduced and p62 levels were increased (Figure 7D–G). Thus, dietary intake of VD_3_ inhibited pro-survival autophagy markers and increased accumulation of p62, showing that autophagic flux was reduced in ER+ mammary tumors of mice supplemented with VD_3_. 

## 4. Discussion

VitD deficiency is widespread in Western populations, particularly among cancer survivors [42,43]. This is a notable concern because low serum VitD levels are associated with an increased risk of a number of diseases including multiple sclerosis [44], cardiovascular diseases [45], depression [46], and some cancers [47]. In breast cancer, results from case–control and prospective cohort studies linking VitD and breast cancer are mixed [48,49]. However, several recent studies report significantly better survival among breast cancer patients with the highest VitD intake and serum 25(OH)D_3_ levels compared with patients deficient for VitD [50,51,52,53]. Two recent epidemiological studies have shown that breast cancer patients supplemented with VD_3_ prior to diagnosis [54], or within 6 months of diagnosis [55], had significantly improved survival and lower rate of breast cancer-specific mortality than patients not receiving supplementation. In contrast, a recent intervention study in the United States failed to find any effect of VitD supplementation on overall or breast cancer mortality [56]. The dose of supplementation in the study (2000 IU VD_3_) may have been too low to sufficiently elevate VitD levels in participants who had low baseline 25(OH)D_3_ levels. This conclusion is supported by a pooled analysis of two randomized trials in which women were supplemented with VD_3_ and a prospective cohort showing that breast cancer risk was significantly lower in those with serum 25(OH)D_3_ higher than 60 ng/mL, compared with women with concentrations below 20 ng/mL [57].

Here, we found that higher VDR expression in ER+ breast cancers predicted significantly longer recurrence-free survival in patients treated with TAM. Since VitD upregulates VDR [28,29,30], also in our study, our findings and findings by others provide convincing evidence that VitD supplementation of deficient patients may improve prognosis among TAM-treated breast cancer patients. 

An apparent explanation as to why higher VDR levels predict better prognosis in patients treated with TAM is that VitD sensitizes breast cancer cells to antiestrogen therapy [58,59,60,61,62,63,64], as also found in the present study. We observed here that calcitriol and its synthetic analog EB1089 increased TAM sensitivity in human LCC1 breast cancer cells. We also found that VitD compounds upregulated VDR and reversed TAM resistance in LCC9 cells. Inhibition of the VDR receptor using siRNA prevented the effects of calcitriol and EB1089 on LCC1 and LCC9 cells. 

We next explored how VitD might sensitize breast cancer cells to TAM. We identified DEGs in ER+ breast cancer patients from four publically available datasets whose tumors contained higher and lower than median levels of VDR. IPA analysis of these DEGs identified a link between UPR/autophagy and tumor VDR status. Specifically, autophagy was the third and UPR was the tenth of the top statistically ranked canonical pathways identified in the analysis. Consequently, we aligned the DEGs with the Autophagy Regulatory Network [35] and identified 30 common autophagy genes. In general, core autophagy proteins were downregulated, supportive of the previous suggestion that VDR suppresses autophagy. Five of these autophagy-related genes were differentially expressed in all four human datasets: ATG12, BCL2, ESR1, PIK3C3, and PRKAG2. These genes likely represent key signaling hubs in autophagy pathways targeted by VitD in breast cancers. 

Using the human breast cancer cell lines LCC1 (antiestrogen sensitive) and LCC9 (antiestrogen resistant) we found that either calcitriol or EB1089 lowered phosphorylation of IRE1α, JNK, and Bcl2, especially in LCC9 cells, suggesting that VitD has the potential to inhibit autophagy in these cells. However, neither calcitriol, EB1089, nor silencing of VDR by siRNA affected Beclin 1, ATG7, or p62 levels in LCC1 or LCC9 cells. These treatments also did not alter growth of the cells. In other breast cancer cell lines, similar doses of calcitriol and EB1089 inhibited cell proliferation and induced autophagy, as assessed by measuring LC3B [19,20,21,26]. There is no evidence that the increase in autophagy seen in these previous studies induced cell survival, but rather calcitriol or EB1089 increased pro-apoptosis autophagy.

TAM activates UPR and autophagy in ER+ breast cancer cells [6]. In TAM-responsive cells, these changes lead to inhibition of cancer cell proliferation and autophagy-induced apoptosis. However, TAM-induced UPR causes activation of pro-survival autophagy in resistant cells. In our study, both calcitriol and EB1089 reduced IRE1α-JNK, pBcl-2, and LC3IIB and increased p62 levels in TAM-treated LCC9 cells. These observations are consistent with an inhibition of TAM-induced UPR and pro-survival autophagy. These effects were mediated via VDR because silencing of VDR blocked the effects of calcitriol and EB1079 on UPR and autophagy markers.

We also explored the effects of VitD on autophagy pathways in an ER+ preclinical mammary cancer model. Tumor incidence and tumor burden were significantly lowered in mice fed a VitD-supplemented diet compared with mice fed a control diet, further supporting a tumor growth inhibitory effect of VitD in ER+ breast cancer. Further, mammary tumors of mice fed a VitD-supplemented diet exhibited higher levels of VDR and suppression of IRE1α-JNK signaling, and markers indicative of pro-survival autophagy. 

## 5. Conclusions

Together the findings obtained using data from breast cancer patients, LCC1 and LCC9 human breast cancer cells, and an animal model suggest that VitD prevents TAM-induced pro-survival autophagy. Whether VitD can be used to prevent or reverse TAM resistance in breast cancer patients remains to be determined. However, the data presented here highlight the potential use of EB1089 or VD3 supplementation with conventional endocrine therapies for the improvement of outcome in ER+ breast cancer patients. 

## Figures and Tables

**Figure 1 nutrients-13-01715-f001:**
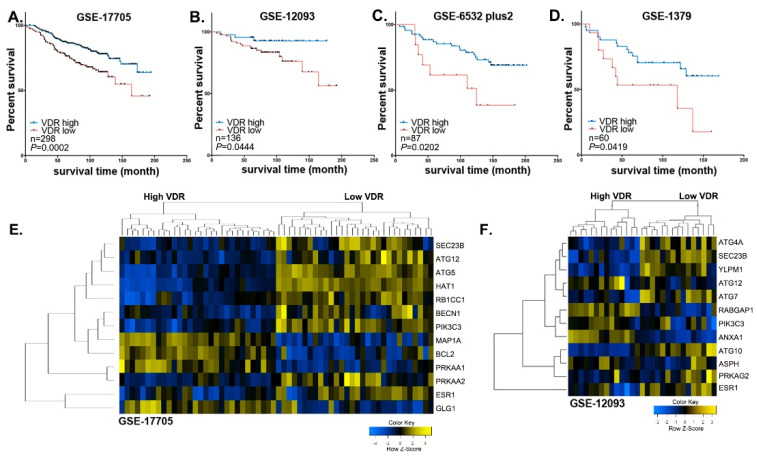
VDR expression and breast cancer survival and autophagy-related gene expression in tamoxifen-treated patients. Recurrence-free survival of ER+ breast cancer patients with above versus below median VDR expression in the following GEO datasets: (**A**) GSE-17705 [31], (**B**) GSE-12093 [32], (**C)** GSE-6532 [33], and (**D**) GSE-1379 [34]. Cutoff optimization was done using R environment for each dataset. Cluster analyses of autophagy-related genes are shown in the heat maps for (**E**) the dataset of GSE-17705; and (**F**) the dataset of GSE-12093.

**Figure 2 nutrients-13-01715-f002:**
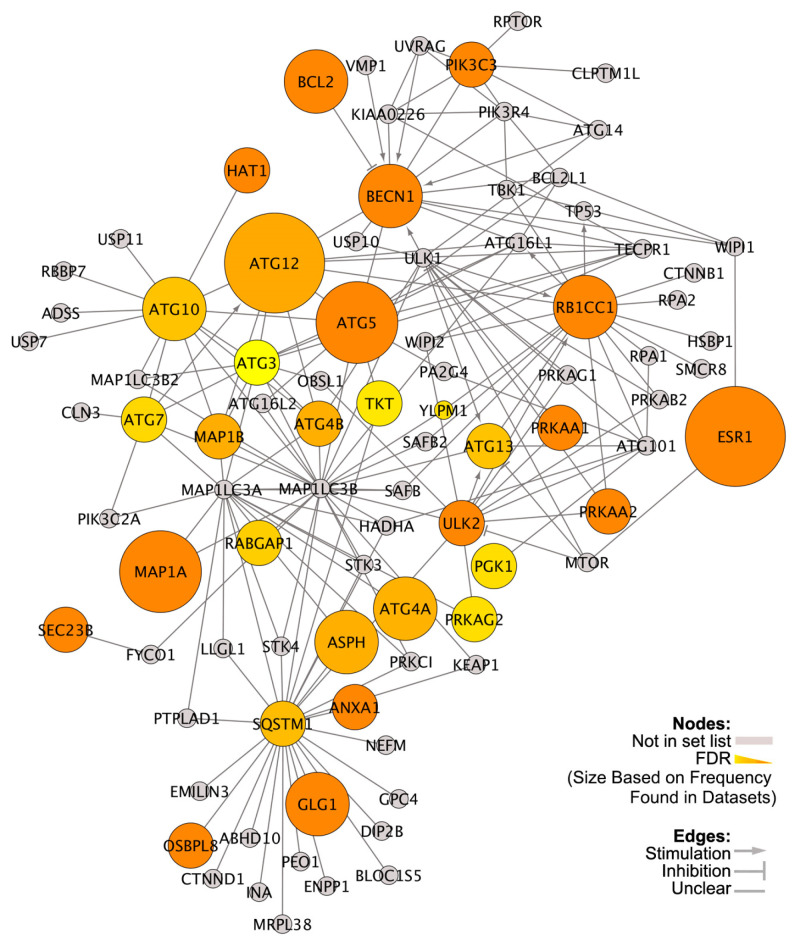
Protein–protein interactions among differentially expressed genes (DEGs) in the Autophagy Regulatory Network (ARN) [35] between breast cancers expressing higher and lower VDR. ER+ mammary tumors in GSE-17705 [31], GSE-12093 [32], GSE-6532 [33] and GSE-1379 [34] datasets were used. Colored circles are DEGs. Node color represents FDR value; the darker the color the smaller the FDR. Node size denotes the frequency of a gene differentially expressed in GEO datasets, e.g., smallest nodes are DEGs in 1 dataset, while largest nodes are DEG in 4 datasets. VDR expression and breast cancer survival and autophagy-related gene expression in tamoxifen-treated patients.

**Figure 3 nutrients-13-01715-f003:**
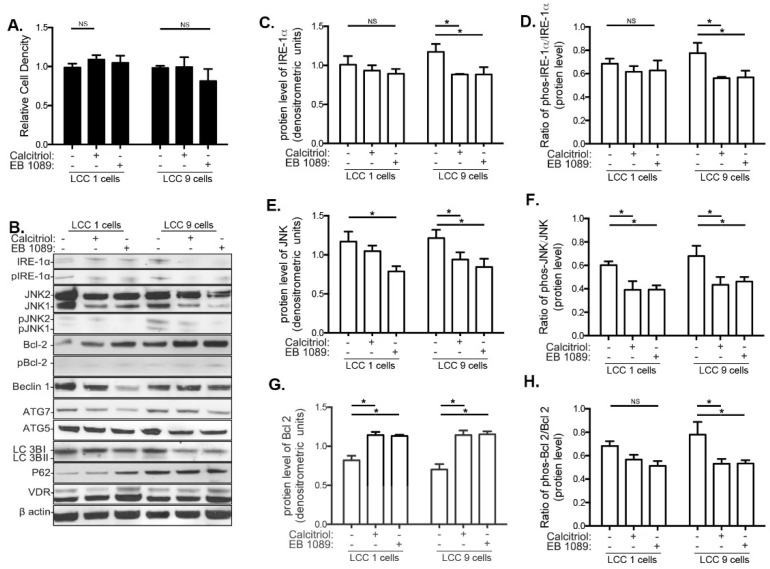
Calcitriol and EB1089 inhibited the activation of the IRE1α-JNK signaling pathway in breast cancer cells. Cells were treated with vehicle (ethanol), calcitriol (100 nM) or EB1089 (100 nM) for 7 days. (**A**) Relative cell density, as determined by crystal violet assay. (**B**) Representative Western blots of the IRE1α-JNK signaling pathway, and quantification of changes in protein expression of three independent biological replicates of (**C**) total IRE1α, (**D**) the ratio of pIRE-1α over total IRE-1α, (**E**) total JNK, (**F**) the ratio of pJNK over total JNK, (**G**) total Bcl-2, and (**H**) the ratio of Bcl-2 over total Bcl-2. β-actin was used as a loading control. * *p* < 0.05, NS: Not significant.

**Figure 4 nutrients-13-01715-f004:**
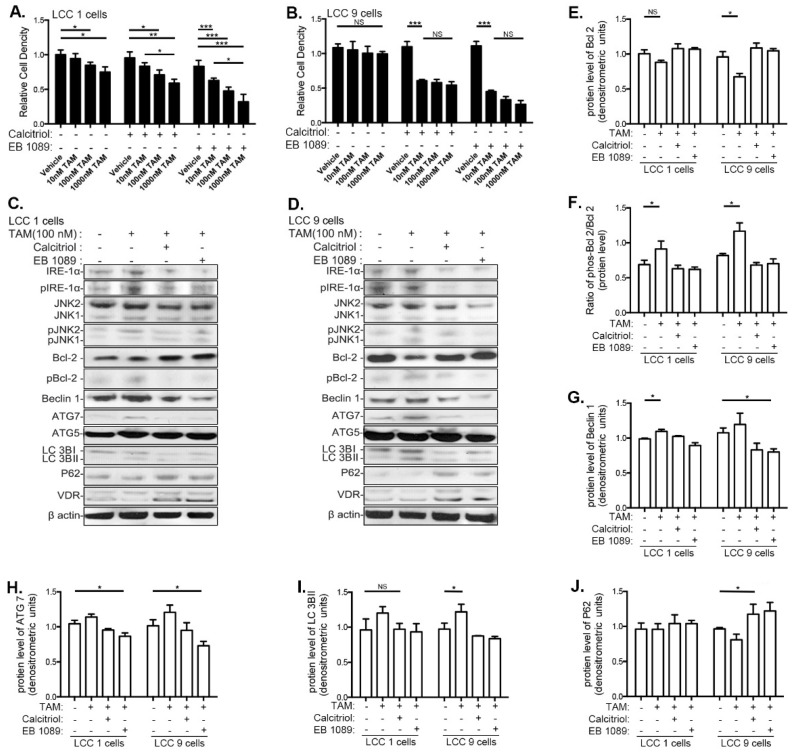
Calcitriol and EB1089 restore responsiveness of breast cancer cells to TAM treatment. Cells were treated with increasing doses of TAM (0–1000 nM), combined with vehicle (ethanol), calcitriol (100 nM) or EB1089 (100 nM) for 3 days, respectively. Relative cell density was determined by crystal violet assay. (**A**) LCC1 cells, (**B**) LCC9 cells. Representative blot from Western blot analyses of UPR and autophagy genes in (**C**) LCC1 cells, (**D**) LCC9 cells. Quantification of protein expression of (**E**) total Bcl-2, (**F**) the ratio of pBcl-2 over total Bcl-2, (**G**) Beclin 1, (**H**) Atg7, (**I**) LC3BII, and (**J**) p62. Quantification based upon three independent biologic replicates. β-actin was used a loading control. * *p* < 0.05; ** *p* < 0.01; *** *p* < 0.001; NS: not significant. Means ± standard error of means are shown.

**Figure 5 nutrients-13-01715-f005:**
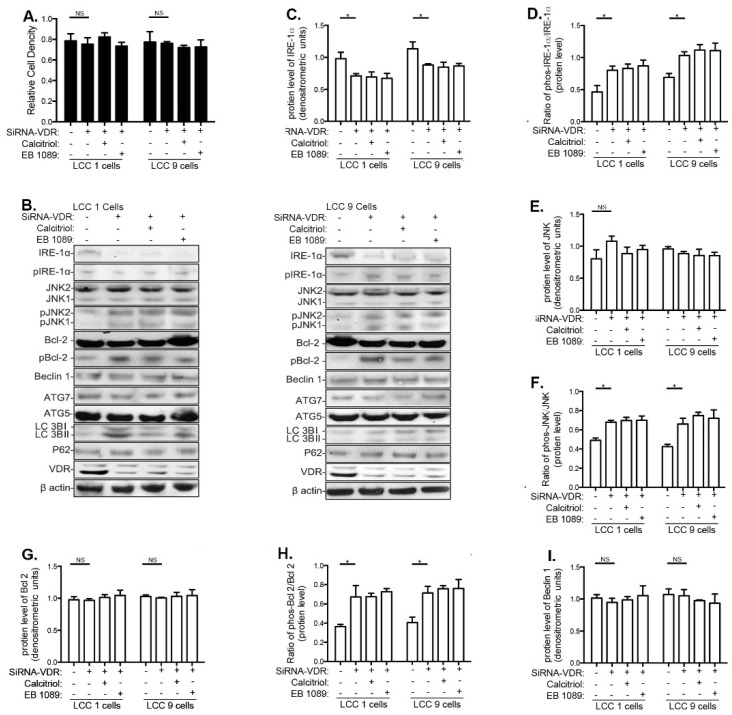
Silencing VDR prevents the activation of the IRE1α-JNK signaling pathway in calcitriol- and EB1089-treated breast cancer cells. (**A**) siRNA- and siRNA-VDR-transfected LCC1 and LCC9 cells were treated with vehicle (ethanol), calcitriol (100 nM) or EB1089 (100 nM) for 3 days post-transfection. Relative cell density was determined by crystal violet assay. (**B**) Representative blot from Western blot analysis of siRNA-VDR-transfected LCC1 and LCC9 cells treated with either calcitriol or EB1089. Quantification of protein expression level of (**C**) total IRE1α, (**D**) the ratio of pIRE-1α over total IRE-1α, (**E**) total JNK, (**F**) the ratio of pJNK over total JNK, (**G**) total Bcl-2, (**H**) the ratio of Bcl-2 over total Bcl-2, and (**I**) Beclin 1. Quantification was based upon three independent biologic replications using β-actin as loading control. * *p* < 0.05. NS: not significant. Means ± standard error of means are shown.

**Figure 6 nutrients-13-01715-f006:**
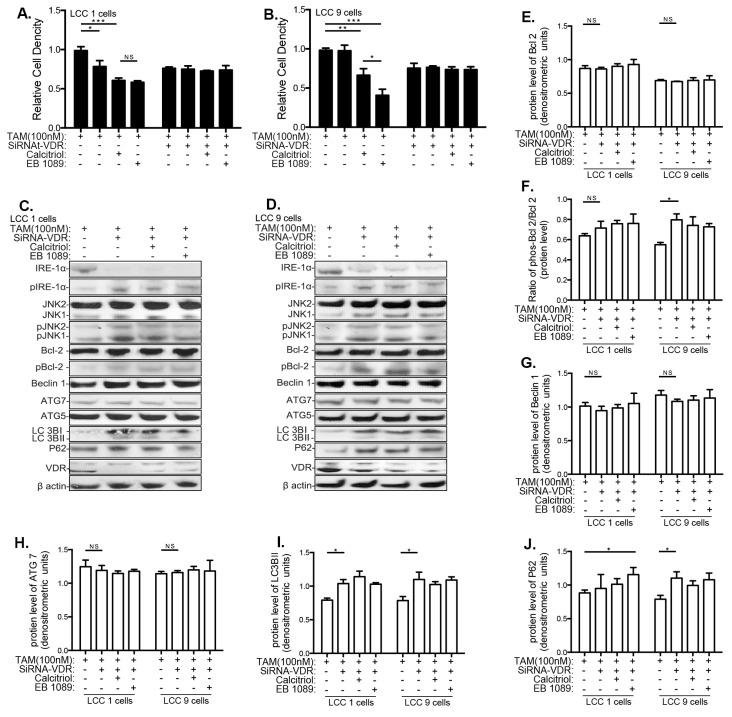
Silencing VDR blocks the response of breast cancer cells to TAM and VitD. siRNA-VDR-transfected LCC1 (**A**) or LCC9 (**B**) cells were treated with TAM (100 nM) plus either vehicle (ethanol), calcitriol (100 nM) or EB1089 (100 nM) for 3 days, respectively. Relative cell density was determined by crystal violet assay. Representative Western blots of autophagy-linked genes in (**C**) LCC1 cells and (**D**) LCC9 cells treated as described above. Quantification of protein expression of (**E**) total Bcl-2, (**F**) the ratio of pBcl-2 over total Bcl-2, (**G**) Beclin 1, (**H**) Atg7, (**I**) LC3BII, and (**J**) p62 based on three independent replications using β-actin as a loading control. TAM treatment. * *p* < 0.05; ** *p* < 0.01, *** *p* < 0.001. NS: not significant. Means ± standard error of means are shown.

**Figure 7 nutrients-13-01715-f007:**
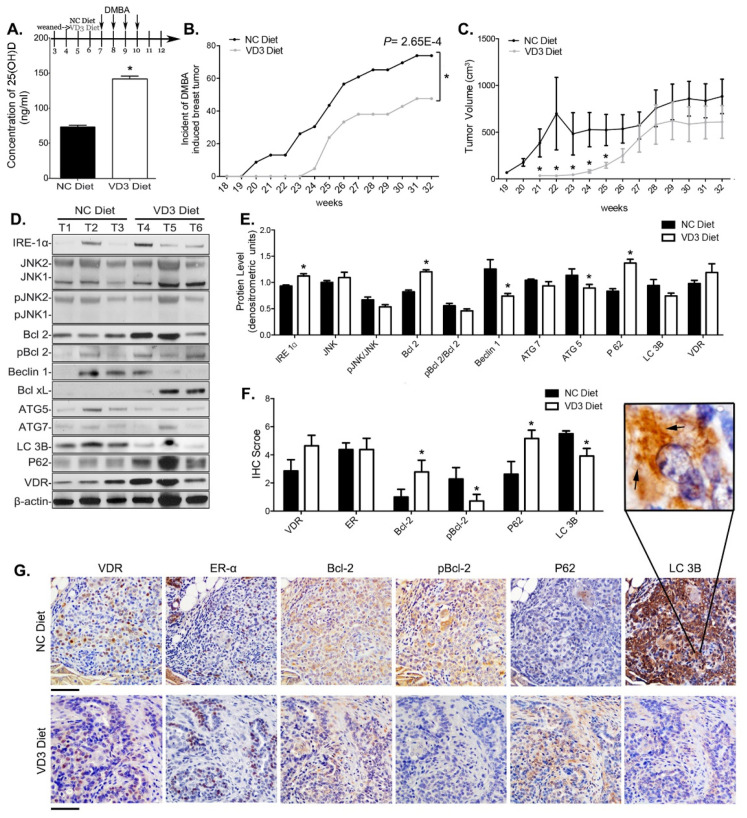
VD3 supplementation reduces mammary cancer risk and inhibits autophagy in mammary tumors. (**A**) Serum 25(OH)D concentration in mice fed AIN93G control diet containing 1K IU VD_3_ (NC diet; *n* = 11) and VD3-supplemented AIN93G diet containing 20K IU VD_3_ (*n* = 11) for 8 weeks. Changes in (**B**) tumor incidence and (**C**) tumor burden (total tumor volume per mouse) in control (*n* = 20) and VD_3_-supplemented diet (*n* = 20) fed mice. (**D**) Western blot analyses of control (*n* = 16) and VD_3_ diet (*n* = 16) fed mouse mammary tumors. (NC diet, *n* = 16; VD_3_ diet, *n* = 16). 3 representative blots from each group are shown with β-actin as loading control. (**E**) Quantification of protein expression levels of the IRE1α-JNK signaling pathway and autophagy. (**F**) Quantification of protein expression levels of VDR, ER-α, Bcl-2, pBcl-2, P62 and LC3B expression by IHC staining (NC diet, *n* = 8; VD_3_ diet, *n* = 8). * *p* < 0.01. (**G**) Representative serial sections of tumor mass with IHC staining of VDR, ER-α, Bcl-2, pBcl-2, P62 and LC3B. Inset is 10×, scale bar: 100 μm.

## Data Availability

The datasets supporting the conclusions of this study are included within the article and its Supplementary files.

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
