# Peer review of "Inhibition of Antiestrogen-Promoted Pro-Survival Autophagy and Tamoxifen Resistance in Breast Cancer through Vitamin D Receptor"

_nutrients, 2021, doi:10.3390/nu13051715_

Round 1

Reviewer 1 Report

This manuscript presents an original study about an interesting subject.

I have just few comments:

-“For 116 each GSE dataset, survival analysis was performed on patients with high (>median) versus low (<median) VDR expression.”

Could you specify if median value is included in >median or <median?

-“We also analyzed RFS in GSE-1456, a dataset of 159 cases of which only 39% were ER+ 235 and treated with TAM [38]. No association between VDR expression and RFS was seen.”
could you specify the P-value?

-"We determined whether VDR levels are predictive of relapse-free survival"  

It is better to define VDR levels as prognostic factors and not predictive factors and use the terminology: 'VDR levels are associated with RFS'

-"Recurrence or relapse-free survival (RFS) in human datasets was defined as 110 the interval from breast surgery to the diagnosis of the first local or distant recurrence"

what about last follow-up for subjects with no relapse? what about deaths? they should be mentioned. please specify how they are considered.

-Please indicate percentages of missing values and median follow-up.

Reviewer 2 Report

Thank you for your contributions.

This is well written and designed study to evaluate VitD supplementation o ER+ Breast cancer to

prevent recurrence. This study results provide the importance of vitaminD  supplementation to the

breast cancer patient survivors.  However, I wonder proper VitD dose or serum level of 25OHD in breast cancer patients according to your study results. Please add if possible.
